# Mutual Two-Way Interactions of Curcumin and Gut Microbiota

**DOI:** 10.3390/ijms21031055

**Published:** 2020-02-05

**Authors:** Ryszard Pluta, Sławomir Januszewski, Marzena Ułamek-Kozioł

**Affiliations:** Laboratory of Ischemic and Neurodegenerative Brain Research, Mossakowski Medical Research Centre, Polish Academy of Sciences, 02-106 Warsaw, Poland; sjanuszewski@imdik.pan.pl (S.J.); mulamek@imdik.pan.pl (M.U.-K.)

**Keywords:** curcumin, gut microbiota, dysbiosis, gut–brain–microbiota axis, herbal medicine, curcumin metabolites, pro-health bacteria, intestinal health

## Abstract

Curcumin, an herbal naturally occurring polyphenol, has recently been proposed for the treatment of neurodegenerative, neurological and cancer diseases due to its pleiotropic effect. Recent studies indicated that dysbiosis is associated with the abovementioned and other diseases, and gut microflora may be a new potential therapeutic target. The new working hypothesis that could explain the curative role of curcumin, despite its limited availability, is that curcumin acts indirectly on the brain, affecting the “gut–brain–microflora axis”, a complex two-way system in which the gut microbiome and its composition, are factors that preserve and determine brain health. It is therefore suspected that curcumin and its metabolites have a direct regulatory effect on gut microflora and vice versa, which may explain the paradox between curcumin’s poor bioavailability and its commonly reported therapeutic effects. Curcumin and its metabolites can have health benefits by eliminating intestinal microflora dysbiosis. In addition, curcumin undergoes enzymatic modifications by bacteria, forming pharmacologically more active metabolites than their parent, curcumin. In this review, we summarize a number of studies that highlight the interaction between curcumin and gut microbiota and vice versa, and we consider the possibility of microbiome-targeted therapies using curcumin, particularly in disease entities currently without causal treatment.

## 1. Introduction

Curcumin is a natural phenolic component derived from the *Curcuma longa* plant and has been used in India to treat inflammation [1,2,3]. Due to its chemical structure (Figure 1), this molecule can be used in several different areas, such as food, textiles and the pharmaceutical industry [1,2,3]. Curcumin is widely used as a spice and dye in food products with a characteristic yellow color; therefore, it is consumed daily [1]. Due to the yellow color, curcumin is systematically used for coloring, e.g., mustard, canned fish and dairy products [1]. It is also used as a cosmetic product, especially for skin. Curcumin is a spice that has recently gained great interest and is widely used in Ayurvedic medicine. Curcumin is a promising compound that is easily available and easy to use in the diet, and it is also safe and affordable. Curcumin is a lipophilic polyphenol that has poor systemic bioavailability and suffers from biotransformation by the human intestinal microflora, to obtain various metabolites that are easily conjugated with glucuronides and O-sulfate conjugates [3]. 

Despite the fact that curcumin has a wide range of therapeutic impacts [1,2] it nevertheless shows extremely low bioavailability [3], which makes it an intriguing substance from a pharmacological point of view, and also hinders its large-scale use in the human clinic [4]. Notwithstanding the abovementioned information on curcumin, currently available data provide evidence that curcumin has antitumor activity, induces neuroprotection and neurogenesis, and can be a new therapeutic agent in both regenerative medicine and neurodegenerative diseases, such as post-ischemic neurodegeneration and Alzheimer’s disease [1,5,6,7,8,9,10,11]. There has been a lot of evidence in recent years confirming the links between the change in the intestinal microflora and many diseases such as cancer, diabetes, autoimmune diseases and neurodegenerative diseases [12,13,14]. An interesting fact is, but it is not surprising, that curcumin is present in high concentrations in the gastrointestinal tract after oral administration. Given the pathogenic links between intestinal microflora and many diseases, current findings could help us interpret the therapeutic advantage of curcumin [15]. In relation to the abovementioned information, we will review current knowledge about mutual two-way interactions between curcumin and intestinal microflora, paying attention to two phenomena: regulation of intestinal microflora by curcumin and biotransformation of curcumin by gut microbiota [15]. In addition, the data review concerns the potential pharmacological uses of curcumin, i.e., identifying metabolites that are more bioavailable and active than parent curcumin, assessing the effect of curcumin regulation on intestinal microflora and its pharmacological activity, and finally presenting strategies for prevention and/or treatment by curcumin based on gut microflora regulations in the light of its clinical safety. Finally, this key review aims to understand at least some of the mechanisms of curcumin action and propose future actions regarding the use of this substance to fight human diseases.

## 2. Search Criteria and Data Collection

The published scientific literatures were searched for in in vivo, in vitro, experimental and clinical studies, and randomized controlled trials of adult human participants, reporting interaction between curcumin and gut microbiota and vice versa. The searches were conducted digitally by using the databases PubMed, MEDLINE, Science Direct, Google Scholar and SCOPUS, to identify peer-reviewed original and review articles in the last two decades (January 1, 2001–January 16, 2020). The search strategy was carried out by using the following keywords: “Curcumin”, “gut microbiota”, “intestinal microflora”, “curcumin and gut microbiota”, “gut microbiota and curcumin”, “curcumin interaction with gut microbiota”, “gut microbiota interaction with curcumin”, “curcumin metabolism”, “curcumin products”, “curcumin metabolites”, “gut microbiota activity”, “gut microbiota products”, “gut microbiota metabolism”, “dysbiosis”, “pro-health bacteria” and “intestinal health”. 

## 3. The Effect of Curcumin on the Gut Microbiota

Curcumin preferentially accumulates in the gastrointestinal tract after oral or intraperitoneal administration, and therefore it is reasonable to conclude that curcumin may have a regulatory effect on the intestinal microflora, including its microbial richness, diversity and composition, which should be involved in its pharmacological action [13]. Curcumin is thought to have a direct regulatory effect on gut microbiota, which may explain the paradox between curcumin’s poor bioavailability and its broadly described pharmacological effects, which cause curcumin’s pharmacology to be clarified [3,13]. Wild-type mice were used to study influence curcumin on gut microbiota in a control and curcumin group. Data indicate that curcumin supplementation tends to reduce the richness and diversity of the intestinal microflora, but without significant differences between the control and curcumin groups. However, curcumin had a significant effect on the number of several bacterial families in the gut, such as *Prevotellaceae, Bacteroidaceae* and *Rikenellaceae* (Table 1) [16]. In other study, curcumin administration significantly changed the ratio of beneficial and pathogenic intestinal microflora by increasing the number of *Bifidobacterium* and *Lactobacilli* and reducing the bacterial load of *Coriobacterales*, *Prevotellaceae*, *Enterococci* and *Enterobacteria* (Table 1). These changes in the intestinal microflora may explain the immune modulation and its antitumor effect in the colon [13]. Other studies concerned the regulatory effect of oral administration of curcumin on the intestinal microflora in mice [14]. A significant reduction in the number of *Prevotellaceae* and *Prevotella* was observed, while the number of *Bacteroides, Alistipes, Bacteroidaceae* and *Rikenellaceae* was significantly increased in the curcumin group (Table 1) [14]. The number of *Prevotella* in patients with colorectal cancer was greater than in feces from cancer-free patients [17]. The role of *Prevotella* in the immune response in gingivitis and peri-implant mucositis has been documented in studies that found a significant relationship between IL-1α and IL-1β levels in crevicular fluid with *Prevotella* colonization [18]. In mice with colitis and colon cancer, increased *Lactobacillales* (Table 1) and reduced *Coriobacterales* after curcumin administration was observed, and curcumin reduced or eliminated colon tumor burden [19]. A review of human and animal studies assessing the relationship between microbiome and colon cancer showed that some bacteria are permanently enhanced, such as *Alistipes*, *Fusobacteria, Porphyromonadaceae, Staphylococcaceae, Coriobacteridae, Methanobacteriales* and *Akkermansia* spp., while others are permanently reduced, such as *Lactobacillus, Bifidobacterium, Ruminococcus, Roseburia, Faecalibacterium* spp. and *Treponema* [12]. In addition, bacterial metabolites like amino acids increased, while butyrate decreased during colon carcinogenesis. Ultimately, evidence suggests that colorectal carcinogenesis is associated with intestinal microbial dysbiosis [12]. Human colon cancer cells differ in their sensitivity to curcumin-induced apoptosis because heat-shock proteins partially prevented apoptosis by inhibiting the release of apoptosis-inducing factor and caspases, although the release of the second caspase activator and cytochrome c did not change [20]. In various colorectal cancer cell lines, curcumin and celecoxib synergistically inhibited both proliferation and induced cell apoptosis via COX-2 and non-COX-2 pathways [21]. Another study showed an effective and safe way to use curcumin loaded in micelles with stearic acid and g-chitosan, in the treatment of colorectal cancer [22]. The positive effects of curcumin in experimental colorectal cancer have led to Phase I clinical trials demonstrating curcumin safety and tolerability in patients with colorectal cancer [23,24]. The overwhelming experimental evidence and completed Phase I clinical trials suggest that curcumin may prove useful in the chemoprevention of colorectal cancer in humans. Positive results of these studies led to the development of Phase II trials for chemoprevention of colon cancer [25]. Another study investigated the effect of nanoparticle curcumin on colitis in mice by modulating intestinal microflora and induction of regulatory T cells [26]. Nanoparticle curcumin inhibited mRNA expression on the mucosa of inflammatory mediators and activation of NF-κB in colon epithelial cells. These effects were accompanied by an increase in butyrate-producing bacteria and fecal butyrate levels [26]. Therefore, the authors concluded that supplementation with curcumin nanoparticles induced a remission of colitis by modulating the structure of the intestinal microflora [26]. Studies on the effects of curcumin in combination with mesalamine on mild to moderate ulcerative colitis and active inflammatory bowel disease have shown that curcumin induced remission of enteritis by modulating intracellular signaling pathways, including anti-inflammatory and immunoregulatory mechanisms [27,28]. Curative effects of curcumin have been reported in atherosclerosis in ApoE/LDLR double knockout mice fed with a Western diet [29]. Curcumin, mixed into the diet, was administered for four months, at a dose of 0.3 mg/day/per mouse. In this model, curcumin inhibited atherosclerosis, measured by both the “en face” and “cross-sectional” methods [29]. Importantly, curcumin did not affect blood cholesterol and triglyceride levels. This explains the effectiveness of curcumin in reversing the effect of a high-fat diet on the composition of the intestinal microflora by shifting it toward slim comparative animals fed a normal diet, which, at the same time, improved the intestinal barrier function [30]. The action of curcumin was studied in animals infected with *Toxoplasma gondii*, it was found that the treated animals showed not only less pro-inflammatory *Enterobacteria* and *Enterococci*, but also higher levels of anti-inflammatory *Lactobacilli* and *Bifidobacteria* (Table 1) [31]. Interestingly, treatment with this compound had a positive effect on the intestinal barrier function, as indicated by a reduced rate of bacterial translocation to the liver, spleen, blood and kidneys [31]. Oral curcumin supplementation alleviates acute small-intestinal inflammation by reducing the Th1-type immune response and prevents bacterial translocation by maintaining the intestinal barrier function. In addition, curcumin administration has been shown to significantly reduce Western-diet-induced blood lipopolysaccharide increase. This study also suggests the positive effect of curcumin on the intestinal barrier and the prevention of metabolic disorders [32]. In this sense, current similar research has shown that curcumin improves intestinal barrier function by modulating intracellular signaling and regulating tight junctions [30]. All of the abovementioned studies suggest that curcumin may prevent metabolic disorders through a mechanism involved in the regulation of the intestinal barrier. These data provide new and potential methods for the prevention and treatment of patients with small-intestinal inflammation [31]. 

Zhang et al. [34] investigated the effect of curcumin supplementation on gut microbiota in ovariectomized rats, using 16S rDNA sequencing [34]. Estrogen deficiency caused by ovariectomy triggered changes in the structure and distribution of gut microflora in rats, and curcumin administration partially reversed changes in intestinal microflora diversity [34]. These authors noted that curcumin caused significant weight loss in ovariectomized animals, probably through modulation of the intestinal microflora. The results indicated that the intestinal microflora of the curcumin-treated rats had a higher level of biodiversity and variability than ovariectomized animals [34]. At the level of the genus, compared to the rats with the ovariectomy group, the numbers of *Incertae sedis*, *Anaerovorax, Helicobacter* and *Anaerotruncus* in the intestines of the sham rats and the numbers of *Serratia, Shewanella, Anaerotruncus, Pseudomonas, Exiguobacterium, Papillibacter* and *Helicobacter* in the intestines of the curcumin group changed significantly [34]. Curcumin may therefore partially reverse changes in the intestinal microflora diversity in animals due to estrogen deficiency after ovariectomy [34].

According to new data on the effect of curcumin supplementation on human intestinal microflora in a double-blind, randomized, placebo-controlled pilot study, the response of the intestinal microflora to treatment was highly personalized, leading to a consistent response of responders and non-responders [33]. These responsive patients had uniform growth in *Clostridium* spp., *Bacteroides* spp., *Citrobacter* spp., *Cronobacter* spp., *Enterobacter* spp., *Enterococcus* spp., *Klebsiella* spp., *Parabacteroides* spp. and *Pseudomonas* spp. (Table 1). Common for these people, the relative number of several *Blautia* species and most of *Ruminococcus* spp. were reduced [33]. This pilot study on healthy people raised many intriguing questions, without answering them, and demonstrated the complexity of human studies, to assess the therapeutic effects of the curcumin herb [33]. 

## 4. The Effect of Gut Microbiota on Curcumin

On the other hand, curcumin is metabolized by the intestinal microflora, which is why it is interesting to detect and identify metabolites of this natural herbal substance. Therefore, studies on curcumin metabolites were conducted, using two models: fecal suspension and a single bacterial strain. Based on existing data, intestinal microbiota plays a key role in the metabolism and biotransformation of curcumin into a number of active metabolites [35,36]. To understand the metabolism of curcumin in the colon, an in vitro model containing human fecal primers was used for research. The results showed that, during 24 h of fermentation, ~24% of curcumin was degraded by human fecal microflora [35]. Three relevant metabolites were found in the cultures used, namely 1-(4-hydroxy-3-methoxyphenyl)-2-propanol, tetrahydrocurcumin (Figure 2) and dihydroferulic acid (Table 2) [35]. Data provided insight into the metabolism of curcumin in the colon, noting that bacterial breakdown products should be considered in further investigations of both bioavailability and curcumin bioactivity [35]. What is more, Lou et al. [37] identified metabolic products of curcumin by human intestinal flora. Twenty-three new metabolites have been identified, and several new pathways for curcumin metabolism through the intestinal microflora have been revealed, namely through reduction, demethylation, acetylation and hydroxylation, or a combination thereof [37]. Regarding metabolism by a single strain of intestinal microflora, demethylation has been found to be an important metabolic pathway of curcuminoids (a mixture of desmethoxycurcumin, curcumin and bisdemethoxycurcumin) through the human intestine of *Blautia* sp. (Table 2) [38]. Curcumin is noted to be converted into two new metabolites, bisdemetylcurcumin and demethylcurcumin, in a cleavage reaction of methyl aryl ether [38]. In addition, the biotransformation of curcuminoids by *Escherichia fergusonii* and two strains of *Escherichia coli* were traced [39]. Three metabolites, namely tetrahydrocurcumin, dihydrocurcumin and ferulic acid, were found in fermentation cultures in all strains used (Figure 2) (Table 2) [40].

In addition, curcumin adduct curcumin-L-cysteine was also detected [40]. Current studies have also investigated the biotransformation of curcumin by *Bacillus megaterium* strain isolated from mouse droppings [41]. Seven metabolites produced by reduction, demethylation, demethoxylation and hydroxylation pathways were found, six of which were presented for the first time [41]. Another study showed that six biologically relevant bacterial strains (*Bifidobacterium pseudocatenulatum, Bifidobacterium longum, Enterococcus faecalis, Escherichia coli, Lactobacillus casei* and *Lactobacillus acidophilus*) are able to metabolize curcumin, with at least a 56% reduction in parent compound measured [42]. There is some evidence regarding the transformation of curcumin by intestinal microflora by a single strain of yeast, fungus or soil with different metabolites [43,44,45]. Regarding the enzymology of biotransformation pathways used by specific microbial species and taxa, very limited information is known to date. Analyses of microorganisms isolated from human feces showed that *Escherichia coli* had the highest curcumin-metabolizing activity through NADPH-dependent curcumin/dihydrocurcumin reductase [46]. 

It was found that the microbial metabolism of curcumin by *Pichia anomala* provided four major metabolites, 5-hydroxy-7-(4-hydroxy-3-methoxyphenyl)-1-(4-hydroxyphenyl)heptan-3-one, 5-hydroxy-1.7-bis(4-hydroxy-3-methoxyphenyl)heptan-3-one, 5-hydroxy-1,7-bis(4-hydroxyphenyl)heptane-3-one and 1,7-bis(4-hydroxy-3-methoxyphenyl)heptane-3,5-diol (Table 2) [36]. Curcumin metabolism has been shown to have phases: In phase I, three metabolites are formed, namely hexahydrocurcumin, tetrahydrocurcumin (Figure 2) and octahydrocurcumin (Table 2) [36]. Curcumin and its phase I metabolites are then conjugated via phase II metabolism, to yield the corresponding metabolites conjugated with glucuronide and O-sulfate [36]. Intestinal microbiota can deconjugate phase II metabolites and convert them back to the corresponding phase I metabolites and some fission products, such as ferulic acid in the colon and cecum [47]. Therefore, curcumin metabolites and degradation products should be considered as mediators of the pharmacological activity of this substance [47]. Glucuronidation is the dominant conjugation pathway of curcumin and its metabolites, and hexahydrocurcumin glucuronide is usually found as the main metabolite of curcumin in body fluids, organs and cells [47]. There is evidence that curcumin metabolites have similar properties and potency to curcumin [48]. Tetrahydrocurcumin is, among others, one of the main metabolites of curcumin (Figure 2); the data indicate that it may also exhibit the same physiological and pharmacological properties as the active form of curcumin in vivo by means of the beta-diketone moiety, as well as phenolic hydroxyl groups [48]. Tetrahydrocurcumin, the main metabolite of curcumin, has been shown to work against neurodegeneration by preventing neuroinflammation and oxidative stress, and it exhibits anticancer effects [49]. These effects may be due to inhibition of significant cytokine release, including interleukin-6 and tumor necrosis factor-α [49]. On the other hand, octahydrocurcumin and hexahydrocurcumin (Figure 2) do not significantly affect the release of cytokines [49]. Lipopolysaccharide-mediated activity of macrophage cells, excess NO production, iNOS and COX-2 activity, as well as NF-κB activation were significantly inhibited by three curcumin metabolites, octahydrocurcumin, hexahydrocurcumin and tetrahydrocurcumin, with tetrahydrocurcumin being the most pharmacologically active (Figure 2) [50]. A detailed analysis showed that the anti-inflammatory mechanism of curcumin and its metabolites can be due to inhibition of IκB-α degradation, which in turn prevented NF-κB translocation into the nucleus [50]. Further mechanistic studies have shown that the use of curcumin-Cu(II) or -Zn(II) complex systems inhibits cellular apoptosis by downregulating the nuclear factor κB pathway and increasing the regulation of the Bcl-2/Bax pathway [51]. To sum up, the presented study showed that the curcumin-Cu(II) or -Zn(II) complex systems show significant neuroprotective effects, suggesting the potential advantage of curcumin as a promising agent against Alzheimer’s disease and post-ischemic neurodegeneration [52] and deserves further research. 

All this evidence suggests the huge potential of curcumin metabolites produced by the intestinal microflora as promising substances for the prevention or treatment of neurodegenerative disorders. Therefore, curcumin metabolites produced by the intestinal microflora and its degradation products should be considered an important source for identifying biologically active molecules. In addition, further research is needed on the enzymology of the metabolic pathways used by the intestinal microflora, to obtain more bioavailable and bioactive curcumin metabolites. Finally, the evidence discussed above paves the way for future research into the development of prophylaxis/treatment based on intestinal microflora with curcumin for many disease entities.

## 5. Conclusions

The two-way interaction between curcumin and intestinal microflora presented in this review includes modulation of the intestinal microflora with curcumin and biotransformation of curcumin by the intestinal microflora, which provides a new basis for understanding the pharmacology of curcumin, and also provides useful tips for future research on this substance. Current knowledge about the two-way interaction of intestinal microflora and curcumin is important to clarify the paradox between curcumin bioavailability and biological activity, and may also point to a new pathway in the prevention and treatment by curcumin of many disease entities in humans that currently have no causal treatment. Curcumin is one of the most commonly studied herbal substances that give hope for the treatment of many diseases, including those in which there is currently no causal treatment. However, it should be strongly emphasized that curcumin is characterized by poor bioavailability and rapid metabolism, which creates problems and raises questions about its effectiveness. In order to respond to emerging doubts and assess the effectiveness of curcumin, as shown by the data presented, it is necessary to address its possible interactions. The data indicate that the interaction of curcumin with intestinal microflora is of fundamental importance, as flora becomes the main player in effectiveness and can pave the way for fulfillment gaps between poor bioavailability and the wide impact of curcumin on health. Based on the evidence presented in the review, the effect of intestinal microflora on curcumin is associated with the production of curcumin active metabolites. Curcumin metabolism can vary from person to person because each person has his or her own microflora composition. Thus, the beneficial effects may be different due to the individual’s different bacterial content, as shown in the review. On the other hand, curcumin may affect the composition of the intestinal microflora, enabling the growth of strains necessary to maintain normal physiological functions of the host. This is the case with neurodegenerative diseases in which intestinal dysbiosis often precedes the appearance of clinical symptoms. Analysis of intestinal microbiota changes in health and disease in the presence of curcumin will identify bacterial strains during and after curcumin metabolism. Data summarized in the review indicate that curcumin itself has a broad spectrum of activity for the development of various pathologies. The role of microflora in enhancing these positive effects can be associated with both the production of more active curcumin metabolites, which can result in better pharmacokinetics, as well as modification of the composition of the intestinal microflora, with a predominance of pro-health intestinal bacteria, such as *Bifidobacterium* and *Lactobacilli*. The results of presented investigations strongly suggest that curcumin may act as a promotion of proliferation, growth and/or survival factors for beneficial members of the intestinal microflora. A number of mechanisms may explain the stimulating effect of curcumin. One of them is the ability of some microorganisms to use polyphenols as substrates. In addition, phenolic compounds have a positive effect on bacteria’s use of nutrients, such as sugars. Additional research of this kind is needed, especially in humans, to accurately understand the mechanisms of modification of the composition of the gut microflora obtained under the influence of curcumin. Future research, which should include a larger human cohort, can clarify whether the responsive microflora that we have described here is representative and whether the less prevalent responses in the data presented can be clearly defined with an increased number of additional participants. Detailed consumption control and/or full diet control in patients can further improve the precision of identifying responsive microbiological taxa to curcumin. Future research should be based on objective measurements of absorption by the host and whether absorption is mainly in the small intestine and/or the large intestine. Future studies should also consider changes in systemic mediators to find out how changes in the intestinal microflora affect biochemical changes in the blood. Research into these characteristics can help establish more precise relationships between intestinal microflora and its potential role as a mediator of curcumin’s health benefits. This will lead to the development of strategies necessary to provide health benefits through microbial modulation and will be the first step to consider new therapeutic uses for curcumin based on intestinal microflora. Modification of the intestinal microflora and its metabolites, as well as curcumin metabolites, will provide new considerations for new therapeutic intervention, especially in diseases currently without causal treatment. 

At present, in animal and clinical studies, toxicological evaluation has shown that curcumin is considered a pharmacologically safe substance at a dose of up to 12 g per day [52]. Currently, extrapolation of animal studies to clinical studies has shown that, in order to obtain beneficial effects in humans, it is recommended to take oral curcumin about 500 mg daily, which means that the daily intake of raw curcumin is about 4 g [52]. However, when using curcumin for therapeutic purposes, it should be stated that it is very unstable in most body fluids, and due to poor water solubility, it is recommended to mix curcumin with milk or oil, to increase its absorption and metabolism [52]. This issue needs to be clarified in future research. Considering the pleotropic effect of curcumin on neurodegenerative diseases, cancers and immune disorders, curcumin is a promising substance for treating disease today without causal treatment [1,2,10,51,52,53,54,55,56,57]. In addition, it is a safe, inexpensive substance, easily accessible and penetrates the blood–brain barrier and neuronal membranes, which is its great advantage [1,2,3,7]. Currently, based on available data, the use of curcumin is contraindicated in people with liver disease, such as cirrhosis, gallstones, gallbladder and biliary obstruction, acute gall colic and obstructive jaundice [7,47,49,52]. Alcoholism or simultaneous alcohol consumption is also a contraindication to curcumin. In addition, curcumin is not recommended for people taking blood thinners, reserpine or nonsteroidal anti-inflammatory drugs, as it may interact with these drugs. In conclusion, the evidence presented in the current literature review of curcumin’s therapeutic potential indicates the potential clinical usefulness of curcumin in the treatment of various diseases currently without known etiology and causal treatment.

## 6. Outlook

Future randomized clinical trials are needed to confirm the mechanisms of curcumin and interactions between curcumin and intestinal microflora and vice versa, and to provide further information on some currently unresolved practical problems, such as therapeutic dose or duration of curcumin use. In addition, these questions can be answered by undertaking large clinical and multicenter studies. To date, data on the use of curcumin as a medicine for the treatment of diseases that are not currently causally treated and associated with dysbiosis, such as brain neurodegeneration and cancer, seem particularly interesting, despite a very limited number of studies. The data showed the association of gut microbiota with curcumin, and vice versa, and underline the need for further research to demonstrate that curcumin is beneficial for disease today without causal treatment. In recent years, curcumin’s reputation in connection with the interaction of intestinal microflora in therapeutic activities has increased significantly. Curcumin gives amazing results in healing activity, as there is no single therapeutic target. In addition, the fact that curcumin, as a natural substance, has more than one target of the drug, allows its versatile use and low risk of inducing resistance to treatment. Recently, attention is drawn to the fact that curcumin treatment and its efficacy have never been demonstrated in a randomized, double-blind, placebo-controlled clinical trial. However, at present, the solution to this problem is facing many difficulties and problems. It should be taken into account that it is very difficult to obtain funding for conducting clinical trials with a substance that will not bring economic benefits in the future, because currently there is unlimited access to it. Another difficult matter is the research assumptions, namely that curcumin cannot be tested in randomized placebo-controlled studies, because clinical trials now require a comparison of the test substance with standard therapy; otherwise, the study will not get approval from the ethics committee. Presumably, this can be bypassed by testing in healthy people, but the results will not completely solve the curative possibilities of curcuma in sick people. Therefore, how do you choose a drug that is now used to treat dysbiosis-related diseases to test the properties of curcumin? There is no doubt that we need to find a solution by looking at in vitro and in vivo animal studies, as well as at preliminary data from the clinic; certainly the next task must be to test curcumin in well-designed and controlled clinical studies. First, future research should focus on the right selection of patients. We hope that the results of future clinical trials will help us better understand the interaction between curcumin and intestinal microbiome, and vice versa. At present, determining the pharmacokinetic profile of curcumin appears to be difficult in many cases, due to low circulatory levels. Therefore, the choice of advanced analytical techniques such as LC–MS/MS should be considered. This technique may extend the scope of research, especially in humans, to accurately understand the modification of microflora composition obtained by curcumin and curcumin metabolites after transformation by intestinal flora. In addition, further research is needed on the enzymology of the metabolic pathways used by the intestinal microflora to obtain more bioavailable and bioactive curcumin metabolites. This may lead to the emergence of strategies needed to provide health benefits through gut microbiota modulation and be the first step to considering novel therapeutic applications for curcumin based on controlling intestinal microflora. 

## Figures and Tables

**Figure 1 ijms-21-01055-f001:**
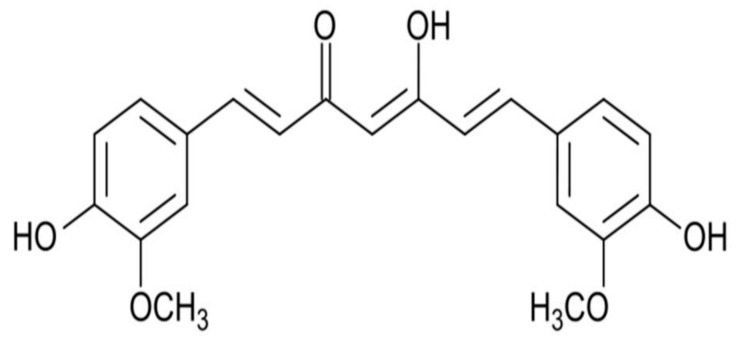
Structure of curcumin.

**Figure 2 ijms-21-01055-f002:**
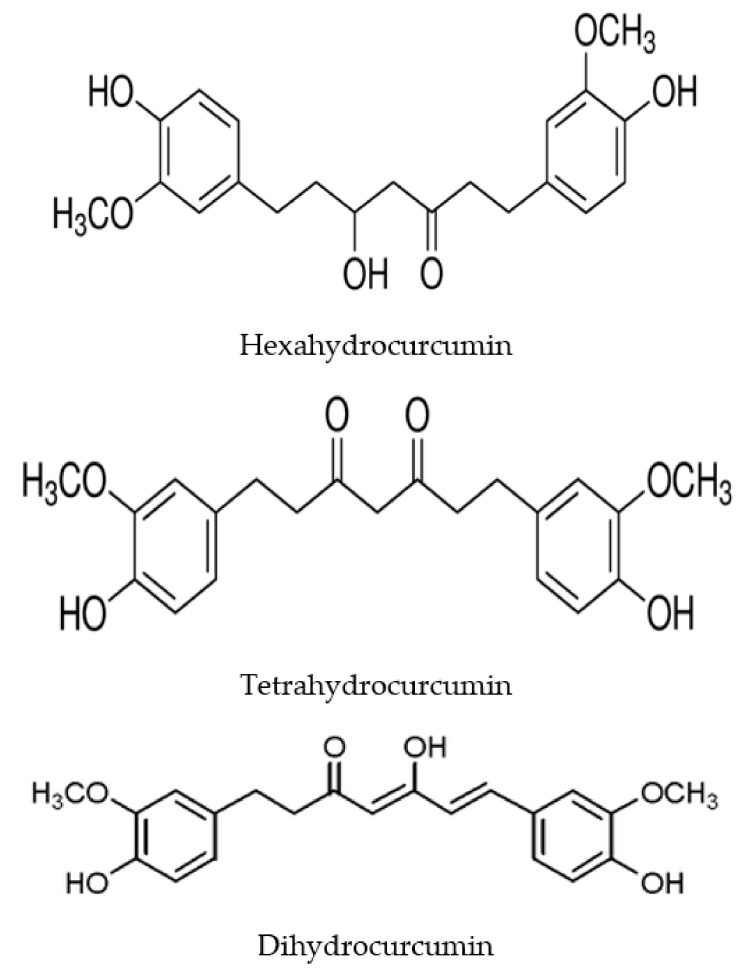
Structure of curcumin main metabolites.

**Table 1 ijms-21-01055-t001:** The effect of curcumin on health-promoting bacteria in the intestinal microflora.

Animal/Human	Pro-Health Bacteria	References
Animal	*Prevotellaceae, Bacteroidaceae*, *Rikenellaceae*	[16]
Animal/Human	*Bifidobacterium*, *Lactobacilli*	[13]
Animal	*Bacteroides, Alistipes, Bacteroidaceae*, *Rikenellaceae*	[14]
Animal	*Lactobacillales*	[19]
Animal	*Lactobacilli*, *Bifidobacteria*	[31]
Human	*Clostridium* spp., *Bacteroides* spp., *Citrobacter* spp., *Cronobacter* spp., *Enterobacter* spp., *Enterococcus* spp., *Klebsiella* spp., *Parabacteroides* spp., *Pseudomonas* spp.	[33]

**Table 2 ijms-21-01055-t002:** The effect of intestinal microflora on the production of curcumin metabolites.

Animal/Human	Material	Metabolites	References
Human	Human fecal primers	1-(4-hydroxy-3-methoxyphenyl)-2-propanol, tetrahydrocurcumin, dihydroferulic acid	[35]
Human	Human fecal starters	Hexahydrocurcumin, tetrahydrocurcumin, octahydrocurcumin	[36]
Human	Intestinal *Blautia* sp.	Bisdemetylcurcumin, demethylcurcumin	[38]
Animal	Intestinal *Escherichia fergusonii*, *Escherichia coli*	Tetrahydrocurcumin, dihydrocurcumin, ferulic acid	[40]
Animal	Intestinal *Pichia anomala*	5-hydroxy-7-(4-hydroxy-3-methoxyphenyl)-1-(4-hydroxyphenyl)heptan-3-one, 5-hydroxy-1.7-bis(4-hydroxy-3-methoxyphenyl)heptan-3-one, 5-hydroxy-1,7-bis(4-hydroxyphenyl)heptane-3-one, 1,7-bis(4-hydroxy-3-methoxyphenyl)heptane-3,5-diol	[36]

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
