# Peer review of "Mutual Two-Way Interactions of Curcumin and Gut Microbiota"

_ijms, 2020, doi:10.3390/ijms21031055_

Round 1
Reviewer 1 Report
The manuscript submitted to IJMS (ijms-707750) is a review concerning the potential interactions of curcumin and gut microbiota. In general, it is well-written and organized paper. However, it is required to provide the structures of curcumin and its metabolites, or even a proposal of transformation pathways of curcumin. There was no comment on the class of compounds which curcumin belong to. In addition, any introduction about sources (plant materials, food) is necessary.
In the section 3. Please provide more information about concentrations (differences) of compound used in the referred studies, e.g. L. 115 „activity of low dose curcumin”. What was the low concentration? More details of studies are needed.
299-303 Please provide any references.More references concerning curcumin are required for this review manuscript.
Author Response
Reviewer 1. The manuscript submitted to IJMS (ijms-707750) is a review concerning the potential interactions of curcumin and gut microbiota. In general, it is well-written and organized paper. However, it is required to provide the structures of curcumin and its metabolites, or even a proposal of transformation pathways of curcumin. There was no comment on the class of compounds which curcumin belong to. In addition, any introduction about sources (plant materials, food) is necessary.
Done. All changes in red.
In the section 3. Please provide more information about concentrations (differences) of compound used in the referred studies, e.g. L. 115 „activity of low dose curcumin”. What was the low concentration? More details of studies are needed.
Done
299-303 Please provide any references.
Done.
More references concerning curcumin are required for this review manuscript.
We added seven.
Reviewer 2 Report
The manuscript is suitable for publication.
Author Response
Reviewer 2.
The manuscript is suitable for publication.
Thanks.
Round 2
Reviewer 1 Report
L. 323 -L. 332
This paragraph also requires references.
Figures 1 and 2: Please change the structures according to the requirements to make them standardized.
Author Response
We did suggested changes.